# Molecular Simulation of CO_2_ and H_2_ Encapsulation in a Nanoscale Porous Liquid

**DOI:** 10.3390/nano13030409

**Published:** 2023-01-19

**Authors:** Pablo Collado, Manuel M. Piñeiro, Martín Pérez-Rodríguez

**Affiliations:** CINBIO, Departamento de Física Aplicada, Universidade de Vigo, 36310 Vigo, Spain

**Keywords:** porous liquid, CO_2_, cryptophane-111, H_2_, molecular dynamics

## Abstract

In this study we analyse from a theoretical perspective the encapsulation of both gaseous H${}_{2}$ and CO${}_{2}$ at different conditions of pressure and temperature in a Type II porous liquid, composed by nanometric scale cryptophane-111 molecules dispersed in dichloromethane, using atomistic molecular dynamics. Gaseous H${}_{2}$ tends to occupy cryptophane–111’s cavities in the early stages of the simulation; however, a remarkably greater selectivity of CO${}_{2}$ adsorption can be seen in the course of the simulation. Calculations were performed at ambient conditions first, and then varying temperature and pressure, obtaining some insight about the different adsorption found in each case. An evaluation of the host molecule cavities accessible volume was also performed, based on the guest that occupies the pore. Finally, a discussion between the different intermolecular host–guest interactions is presented, justifying the different selectivity obtained in the molecular simulation calculations. From the results obtained, the feasibility of a renewable separation and storage method for CO${}_{2}$ using these nanometric scale porous liquids is pointed out.

## 1. Introduction

The definition of porous liquids was introduced by O’Reilly et al. [1], referring to fluid media with permanent porosity, which is achieved through the suspension of large molecules whose geometry include voids acting as intrinsic permanent pores. This breakthrough conception opened a path towards the development of innovative and tunable techniques in applications as gas separation and storage. In this sense, porous liquids represent a good compromise between the great variety of known solid porous materials used for gas selective adsorption, and liquids, which are easier to store and handle at larger scale. Recent progress in the development and characterisation of porous materials has been reviewed in several excellent compilations [2,3,4]. However, few research works have been devoted so far to the topic of porous liquids, due of its novelty. Among the first studies reporting experimentally synthesised porous liquids, Giri et al. [5] created a cage molecule by ether aggregated imines with a big solvent, such as 15-Crown-5, sterically hindered from entering in the cavity of this molecule, but fluent enough to allow the flow of molecules in the system, and Zhang et al. [6] designed and fabricated a porous liquid based on hollow silica spheres.

Porous liquids are currently classified in three groups: Type I, II and III [7]. Type I are composed by liquids where each individual molecule presents a rigid and permanent cavity, meaning that they cannot be auto-filled. This feature makes them the harder to synthesise of the three groups. They present high melting points and a tendency to crystallise outside certain ideal range conditions [8]. Type II are conformed by rigid cage molecules and a sterically hindered solvent which cannot enter the cage cavities. This kind of porous liquid is the easier to prepare, due to this cage molecule not being in its liquid form, though it means it requires a sufficiently soluble solvent to be in and be fluent. Type III porous liquids consist of metal organic frameworks (MOFs) suspended in a sterically hindered solvent. These latter are easier to synthesise, but they may present phase separation and generate precipitation due to the different nature of the pores and its distribution on the MOF, lowering the overall porosity. To avoid this, some studies have considered the use of nanoparticles or nanocrystals despite their big size and the consequent stability issues [9]. The possibility of developing a renewable route to adsorb CO${}_{2}$ in mild conditions has led to several studies of its interaction with porous liquids [10,11,12,13,14]. However, there is still scarce knowledge about the adsorption of H${}_{2}$ in porous liquids, except the study of Oltean et al. [15] where its interaction with cryptophane-111 (C-111) in presence of CH${}_{4}$ is described. In this study we analyse through molecular simulations the process of encapsulation of CO${}_{2}$ and H${}_{2}$ in a porous liquid, at diverse conditions of temperature and pressure. We have considered the case study of a Type II porous liquid, composed by the smallest molecule from the cryptophane family, namely cryptophane-111. This molecule was first discussed by Fogarty et al. [16], and since then it has been intensively studied for his remarkable complexation with Xe [16,17,18]. Cryptophanes are aromatic molecules defined by three symmetric folds of cyclotribencylene that are known to exist in a crown conformation keeping a cavity with stable geometry. They present a great variety of structures and properties due to its stereoisomerism, the nature of their bonded moieties and their functional groups. The possibility of enclathrating an external molecule in their cavity depends on these two latter characteristics, as they define the size and flexibility of the cage geometry [19]. In Figure 1, we present the chemical skeleton of C-111, and a representation of the molecular model built for the simulations performed in this work.

The narrow hollows of C-111 yield a perfect molecule to study the encapsulation/ separation of small molecules. Bearing this is mind, we decided to compare the behaviour of CO${}_{2}$ and H${}_{2}$ in the same porous liquid to analyse their selectivity considering the occupation of the cavities. As for the solvent, DCM was selected because of the solubility of C-111 in it. The cavity volume of C-111 molecules ranges from 32 Å${}^{3}$ to 72 Å${}^{3}$ (in its most expanded state) and, despite DCM presenting a volume inside that interval, it cannot get through the entrance of the cavity, as indicated by Buffeteau et al. [20]. The election of a Type II porous liquid was motivated as well by the easier approach to recreate it in a molecular simulation. The possibility, as mentioned, of diverse precipitations for Type III porous liquid, or the complex structures of Type I porous liquids could lead to ineffective simulations, while the selected case study represents a good starting point before studying more complex scenarios.

The mild equilibrium thermodynamic conditions of PLs in general, and that of the studied case studied here in particular, and the remarkable selectivity in the encapsulation of different guest molecules, have opened innovative perspectives in their used as separation and capture media for different gases. This includes, for instance, the perspective of separating greenhouse gases from industrial flue gases, playing a role in the policies against global warming and climate change. The results presented in this work, using molecular dynamics for the simplest molecule of the cryptophane family, open new perspectives of further analysis of more complex molecular geometries, including even the possibility of adding grafted functional groups to enhance selectivity. The theoretical approach used in this work shows remarkable ability, as will be demonstrated in the following, to capture the physico-chemical subtleties of this PL selective adsorption process.

## 2. Computational Methods

We retrieved a set of molecular structures from different studies. Few works concerning C-111 have been published so far and, thus, building the topology from a previous topology model of the molecule was the initial step. We used the pdb archive of the NOVMUI deposit number 1025559 molecule for the C-111 [17]. This molecule was in the presence of DCM and Xenon, which were eliminated from the file. The topology of cryptophane-111 was built using the Automated Topology Builder (ATB) v.3.0 repository [21]. The molecular model for DCM proposed by Stroet et al. [22] was used, which was obtained also from ATB. Many CO${}_{2}$ molecular models are available in literature developed for their use in molecular simulations. We decided to use the TraPPE [23] molecular model because it has repeatedly shown excellent performance. In previous studies, we used this CO${}_{2}$ molecular model in the study of solid–liquid equilibrium [24], the description of CO${}_{2}$ hydrates [25], or CO${}_{2}$ organic clathrates [26], in all cases yielding excellent performance in systems including solid phases, a key feature also in this study. A two-atom-type hydrogen molecule model has been also used: one atom type is for each hydrogen atom and the other represents a virtual centre, as described in previous works [27,28,29]. For the sake of comparison, the different molecular models used are shown in Figure 2.

All molecular dynamics simulations were run using GROMACS [30,31,32,33,34,35,36,37], 10 ns of simulation with a 1 fs timestep. An initial energy minimisation was performed with the conjugated gradient method to avoid significant molecular overlapping. Electrostatic long range corrections were handled using the particle mesh Ewald technique [38], with 1 nm radius, 4th order and a Fourier spacing of 0.1 nm. The dispersive van der Waals cut-off radius was set to 1 nm, and corrections of long-range dispersion for energy and pressure were applied. A Nose–Hoover [39,40] thermostat with 2 ps coupling constant was used to keep temperature constant, and Parrinello–Rahman [41,42] barostat with 4 ps coupling constant was used to set pressure value. All simulations were built from randomly generated initial states. The final simulation states and their trajectories were analysed using VMD [43,44]. The volume of the cavities and of diverse aspects of the nature of the molecules and the interaction among them were studied using CAVER [45] and PyMol [46].

The simulation box consisted of seven molecules of C-111 and 300 DCM molecules. As for both CO${}_{2}$ and H${}_{2}$, 28 molecules were simulated in each case. Because DCM volume is smaller than the maximum cavity volume of the C-111, special care was taken during the initial setup simulation to avoid the accidental insertion of a DCM molecule inside a C-111 cavity. Then, this system of C-111 and DCM constitutes a porous liquid, because the C-111 porosity is not accessible to the solvent, as the access gate to the cavity is smaller than the DCM dynamic size.

The molecular simulations were organised as follows. The simulation box was built with the composition explained, by inserting first the C-111 molecules, and after that the DCM solvent molecules, by preventing accidental insertion of DCM inside the C-111 cavities due to the reasons explained. Finally, the CO${}_{2}$ and H${}_{2}$ molecules were inserted, also at random, and in this case the inner C-111 porosity was not blocked, so any of these two molecule types might appear inside a C-111 cavity at the beginning of the simulation. In order to check the reproducibility of the simulation, four independent simulation boxes were built using this method, and they were used as replicas for the same simulation condition in each test performed. These four system replicas are labelled S1 to S4 in Figure 3, which shows a scheme of the different simulations carried out. After the energy minimisation step, NpT molecular dynamics was run at 1 bar and 300 K for each system, these simulations being termed as ST. The next step was to run an additional set of simulations trying to evaluate the effect of temperature, by increasing its value to 312 K (HT series). Finally, the ST box was used as starting point to evaluate the pressure effect on the system by performing a low pressure (LP, 0.005 bar) and a high pressure (HP, 100 bar) calculation for each of the four systems tested.

The cavity occupation for every simulation was recorded at the beginning, middle and end of the simulation to compare the evolution for each species at the different conditions explored.

## 3. Results and Discussion

### 3.1. ST Simulations

The conditions for the series of ST simulations were 1 bar and 300 K as described before. The simulation box for the four replicas of the simulation performed were built as described previously by random insertion of the molecules. Table 1 summarises the results obtained in the four runs, that will be explained in detail in the following. In the initial simulation box of the first simulation, denoted as S1ST, no CO${}_{2}$ was initially placed in a cavity, but three H${}_{2}$ molecules were inside the C-111 cavities. At the middle of the simulation, three CO${}_{2}$ had entered in different cavities, while H${}_{2}$ molecules had got out of their cavities and three other different H${}_{2}$ residues had taken their place, two of them eventually in the same cavity, as seen in Figure 4. By the end of the simulation, a new CO${}_{2}$ residue had occupied another cavity, while all previously adsorbed H${}_{2}$ molecules had got out their cavities and a new molecule had occupied one. Thus, starting from a situation where only one CO${}_{2}$ and three H${}_{2}$ were placed inside a cavity, at the end of the simulation run, four CO${}_{2}$ and one H${}_{2}$ occupied four out of seven C-111 available cavities. This information is gathered in Table 1 under the heading S1ST. The main conclusion obtained from this run is that CO${}_{2}$ molecules are adsorbed inside the C-111 cavities, remaining there once enclathrated. On the other hand, H${}_{2}$ molecules may go in and out of the cavities, with low retention times.

The same trend occurs for simulations S2ST to S4ST. In all cases, notwithstanding the initial C-111 cavities occupation, the dissolved molecules show identical behaviour, with CO${}_{2}$ entering the cavities and being trapped inside, while the less interacting H${}_{2}$ are able to easily go in and out of the pores. Figure 5 captures the curious lapse during a simulation run, where a CO${}_{2}$ molecule kicks out an H${}_{2}$ from a cavity and occupies its place, in a process with a duration of 75 ps.

We can see how the repetition of H${}_{2}$ occupancy is casual, as it flows in and out of the cavity of the C-111, but CO${}_{2}$ does not leave a cavity when it gets in. If we compare the number of occupations of H${}_{2}$ and CO${}_{2}$ during the simulation, we can observe that H${}_{2}$ tends to dominate the occupation of cavities during early phases, but as the simulation goes by, CO${}_{2}$ molecules have a greater selectivity towards cavity occupation and this will be appreciated better in the rest of the study. We can also explain this selectivity visually, examining in Figure 5 how a CO${}_{2}$ replaces a H${}_{2}$ molecule initially located inside a cavity, and after that stays in the cavity for the rest of the simulation.

Therefore, the conclusion of this series of ST simulations is that, starting from a configuration where most of the C-111 cavities were empty at the beginning, resulting from a random insertion of the molecules, during the simulation, a stable uptake of CO${}_{2}$ in the porosity is very relevant, while H${}_{2}$ is able to go in and out of the cavities, showing no preference for staying adsorbed.

### 3.2. LP Simulations

After having finished the initial ST tests, the final state of each system replica was used as starting point for other simulations, where the thermodynamic conditions were modified by changing temperature or pressure. In the next set of calculations, pressure was reduced from 1 bar to 0.005 bar to run LP simulations. Table 2 presents the account of the simulation results for the four replicas run, showing the information equivalent to that given for the previous SP simulations. In these conditions again the same behaviour is reproduced. In general terms CO${}_{2}$ molecules may enter but not leave the C-111 pores, but in the case of simulation S2LP we can see an exception to this trend. In a process that occurs during a total lapse of 15 ps, we can see how a CO${}_{2}$ molecule gets replaced by another CO${}_{2}$ molecule, as displayed in Figure 6. This is an interesting phenomenon, that produces a transitory and unstable double CO${}_{2}$ occupancy within the cavity, but represents an oddity if compared with the usual adsorption behavior of CO${}_{2}$ in this porous liquid. This double CO${}_{2}$ occupancy is definitely a rare event for this system, due to the instability of this high load occupation mode. The consideration of higher inner volume molecules of the cryptophane chemical family might lead to a higher possibility of multiple guest occupancy modes, yielding higher storage ability. This is an aspect of remarkable interest considering the possibility of practical CO${}_{2}$ separation and capture applications, but it is clearly beyond the scope of this study, devoted only to the case of C-111 porous liquid.

We can observe again the tendency of CO${}_{2}$ to occupy cavities as the simulation progresses. The H${}_{2}$ keeps being less selective than CO${}_{2}$, and tends to flow over the cavities. As seen in Figure 6, the replacement of a CO${}_{2}$ by another CO${}_{2}$ is possible under low pressure conditions, and it could mean that its interaction and stabilisation inside the cavity is disturbed.

### 3.3. HT Simulations

In this case, starting again from the final SC simulation configurations, temperature was increased from 300 K to 312 K, at 1 bar, and results are summarised in Table 3. An overview of the comparison with the previous ST results allows to identify a higher final occupancy rate of the available cavities in these higher temperature HT simulations. The distinctive adsorption behaviour for CO${}_{2}$ and H${}_{2}$ observed in the previous cases also appears now, but the number of adsorbed CO${}_{2}$ molecules is clearly higher in the present case.

### 3.4. HP Simulations

In this case, pressure was set at 100 bar, keeping the initial temperature of 300K. Results are gathered in Table 4. In these conditions, a high cavity occupancy ratio can also be pointed out, resembling the previous HT cases. The effect of temperature and pressure seems to be significant in the adsorbing pattern of this porous liquid, due to the different interactions of both guest molecules inside the C-111 pore cavity.

### 3.5. ST2 Simulations

A common feature observed in HT, HP and LP simulations is that the occupancy rate of the C-111 cavities increased in all cases if compared with the initial state, which was common to every case and corresponding to the final micro-state obtained from the initial ST conditions simulations. Let us recall that ST simulations started from a situation where most cavities were vacant, while HT, HP and LP started from a system where several of the cavities hosted a guest molecule. This means that a direct comparison between the four simulation sets could lead to unfair conclusions. For this reason we decided, for the sake of equal terms comparison, to extend ST simulations, keeping the original 300 K and 1 bar conditions, during an additional 20 ns. This means that the difference between ST and ST2 simulations is that the first set initiated from a rather empty porous liquid, while ST2 simulation were initiated with a system displaying higher occupation.

The new results are gathered in Table 5, and show interesting conclusions. The most evident result is that cavity occupation is increased if compared with ST simulations, reaching almost full C-111 occupancy in all cases. This process advances following the same trends observed in the previous cases; CO${}_{2}$ molecules find their way to enter the cavities and stay adsorbed there, while H${}_{2}$ molecules enter and go out of the cavities with reduced residence times. The possibility of following the trajectory of each individual molecules allows to determine precisely their evolution, identifying, for instance, which of the H${}_{2}$ molecules enter or leave each of the C-111 cavities. On average, and comparing the different runs, the number of captured CO${}_{2}$ molecules is larger at the end of the calculations, underlining the porous liquid selectivity for these molecules.

Additionally, during these runs, another double occupation event has been identified, with a cavity hosting in this case one CO${}_{2}$ and one H${}_{2}$ molecules, as shown in Figure 7. It is also worth commenting that during the run S1ST, all cavities were eventually occupied by CO${}_{2}$ guests, evidencing the high selectivity for this molecule in the system tested. This fact allows us to infer the potential use of this porous liquid to selectively capture and separate an individual component of a certain gas mixture, caused by the different interaction affinity with the inner C-111 cavities. It is the only full CO${}_{2}$ occupancy case detected, but the tendency of CO${}_{2}$ to replace H${}_{2}$ from the cavities, and the stability of their adsorption, demonstrated by the very reduced number of events of one CO${}_{2}$ leaving a cavity detected during the whole set of simulations performed, supports this conclusion.

### 3.6. Cavity Occupation

Summarising the results of the different simulation runs performed, the average CO${}_{2}$ cavity occupancy was determined for the different thermodynamic conditions explored. This calculation was not performed for the case of H${}_{2}$, due to the stated flowing nature of its adsorption within the cavities. Let us recall that the total number of C-111 molecules in the system was seven. If we denote by ${\bar{N}}_{CO_{2}}$ the number of CO${}_{2}$ molecules adsorbed within the C-111 pores at the end of each simulation set, the results obtained for this calculation were ${\bar{N}}_{CO_{2}}\left(LP\right)=3.5\pm 1.3$; ${\bar{N}}_{CO_{2}}\left(HT\right)=3.5\pm 1.0$; ${\bar{N}}_{CO_{2}}\left(HP\right)=3.5\pm 1.3$; ${\bar{N}}_{CO_{2}}\left(ST2\right)=5.0\pm 1.6$. Double occupancy events, being clearly rare, were not considered in this calculation. These results show that it is around the initial conditions of 300 K and 1 bar that the porous liquid CO${}_{2}$ selectivity and uptake yielded optimal values.

### 3.7. Cavity Volume

C-111 cavity volume ranges from 32 Å${}^{3}$ to 72 Å${}^{3}$ as mentioned before. Using the final snapshots of the simulations performed, a comparative study of the cavities volumes was performed using Caver [45] software. As a visual representation, Figure 8 shows the available empty space within a C-111 cavity using this method. It is interesting to determine whether the presence of a guest produces a significant distortion of the cavity, and with the aim to check this, the inner cavity volumes were computed for each of the simulation series, taking into account separately the empty cavities, and those occupied by the different guests. Results are gathered in Table 6, showing that, despite some minor differences, the C-111 cavities fluctuate around an average values, and the presence of a somewhat larger guest (in the case of CO${}_{2}$) does not produce a relevant increase in the accessible volume, thus pointing to a remarkable structural rigidity of the C-111 cavities, whose pore size distribution is not affected by the occupancy, nor by the guest nature.

### 3.8. CO${}_{2}$ and H${}_{2}$ Interaction with C-111

The selectivity of the porous liquid for CO${}_{2}$ over H${}_{2}$s may be comprehensively interpreted by studying the interactions of these guest molecules inside the C-111 cavity. With this objective, we analysed the different configurations obtained during the performed simulations where CO${}_{2}$, H${}_{2}$, two CO${}_{2}$, two H${}_{2}$ or one CO${}_{2}$ and one H${}_{2}$ appeared inside one of the C-111 cavities. Pymol was used in these configurations to search for polar contacts among these molecules. Figure 9 and Figure 10 show that CO${}_{2}$ forms a type of polar interaction inside the C-111 cavity with one of the C-111 ether groups, either when it is alone, corresponding to the former figure, or when sharing the cavity with an H${}_{2}$ molecule, in the latter case. The absence of H eliminates the possibility of hydrogen bond interaction inside the cavity, but a π–π interaction is feasible in this case. Jie Yin et al. [47] studied the absorption of CO${}_{2}$ and CHCl${}_{3}$ in a porous liquid is studied, and observed as well this π–π interaction between CO${}_{2}$ and the structure of aromatic rings. In this case, they analysed how one oxygen atom of CO${}_{2}$ interacted with the aromatic ring, while in our case just the oxygen interacts with an ether group outside the ring. in another study, Alula et al. [48] explain the adsorption of CO${}_{2}$ in some Poly(ether-block-amide) Co-polymer membranes, concluding that due to the dipolar and quadrupolar interaction there is a non-covalent polar nature bond between the guest CO${}_{2}$ and the cage and this is, in our opinion, the same process that we can identify in our simulation. The other possible π–π interaction, with the aromatic rings, could appear at the same time but it has not been detected in our study, or maybe the charge distribution generated by this dipole–quadrupole interaction makes it not possible to form this π–π bond.

No noticeable interaction has been identified between H${}_{2}$ and CO${}_{2}$ inside the cavity, either for H${}_{2}$ or the C-111. However, we can observe mutual interaction when there are two CO${}_{2}$ inside the cavity, as seen in Figure 10. This representation denotes polar interaction between both CO${}_{2}$ sharing a cavity. In a previous study, Pérez-Rodríguez et al. [49] analysed the interaction of CO${}_{2}$ molecules in adjacent cells in hydrates, finding two stables conformations; the lowest energy corresponds to the case where CO${}_{2}$ are mutually oriented in a T shape, and the second is where both guests are parallel. These two mutual orientations minimise the quadrupolar interaction between them. In Figure 11, we can identify this T shape, and also the cited polar interaction with the ether groups in the cavity. This mutual orientation is quite stable but, as displayed also in Figure 11, the breakup of this conformation provokes one of the CO${}_{2}$ to abandon the cavity, this process corresponding to the removal of one CO${}_{2}$ molecule in the S2LP simulation, as detailed earlier. This may occur due to a charge alteration in the surrounding of one CO${}_{2}$ molecule, inducing a rotation and loss of stability within the cavity, but more cases should be studied to validate this argument.

## 4. Conclusions

This study is a contribution in the theoretical modelling of novel porous liquids, and characterises a potential method for safe and easy CO${}_{2}$ adsorption. As seen in the simulations performed, CO${}_{2}$ adsorption is favoured inside C-111 cavities in a DCM solvent. CO${}_{2}$ tends to occupy empty pores or replace less interacting H${}_{2}$ molecules inside them, and its selectivity improves with time. In our calculations, atmospheric conditions also improve the pore CO${}_{2}$ uptake, reaching a possible full occupation. Two CO${}_{2}$ molecules can stay for a brief period of time in the same pore but this situation is not stable, and one of them soon leaves the cavity. We have also observed the capacity of H${}_{2}$ to enter and leave the pores repeatedly, presenting little selectivity, that may appear higher in early stages of the simulation, but showing a tendency to flow through the pores. The presence of guest molecules inside the C-111 pores, or the different thermodynamic conditions tested, do not affect the accessible inner volume of the cavity. We have also observed quadrupole–dipole interactions between the guest CO${}_{2}$ molecule and the ether group inside the cavity, explaining the stabilisation of this molecule inside the pore, unlike the case of H${}_{2}$ molecules which do not present any noticeable interaction. We also discussed the interaction of two CO${}_{2}$ when they meet inside the cavity.

These results highlight the importance of porous liquid as a potential method to separate and capture gases due to the effect of the intrinsic porosity and the selectivity among different potential guests adsorbed. The possibility of using them as a renewable option of capturing greenhouse gases as CO${}_{2}$ opens a path for significant and environmentally friendly future applications of porous liquids.

## Figures and Tables

**Figure 1 nanomaterials-13-00409-f001:**
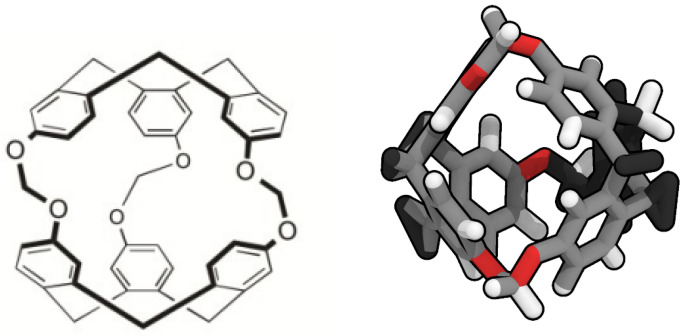
Chemical structure (**left**) [20] and molecular simulation model (**right**) of cryptophane-111.

**Figure 2 nanomaterials-13-00409-f002:**
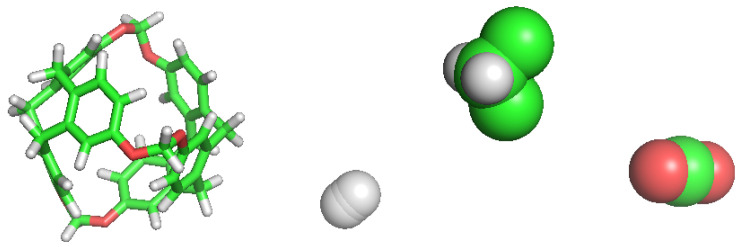
Simulations models for each molecule used in this work. From left to right: C-111, H${}_{2}$, DCM and CO${}_{2}$.

**Figure 3 nanomaterials-13-00409-f003:**
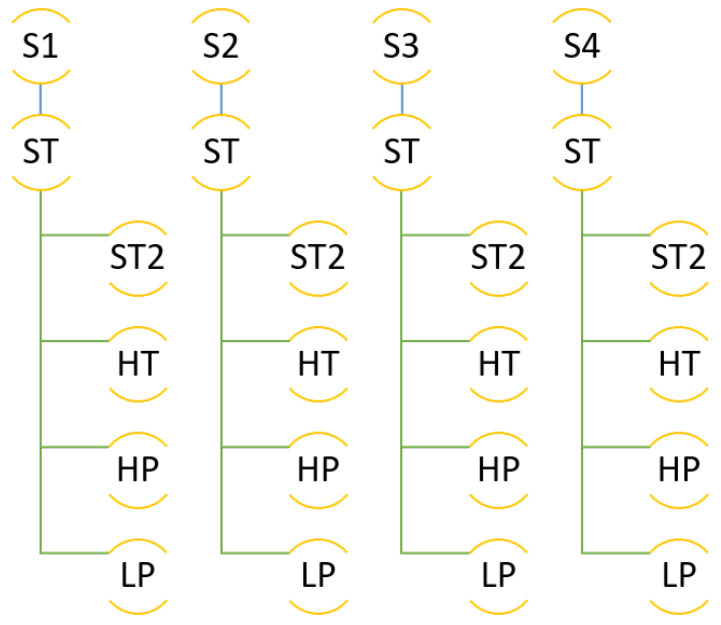
Scheme of the different simulations performed.

**Figure 4 nanomaterials-13-00409-f004:**
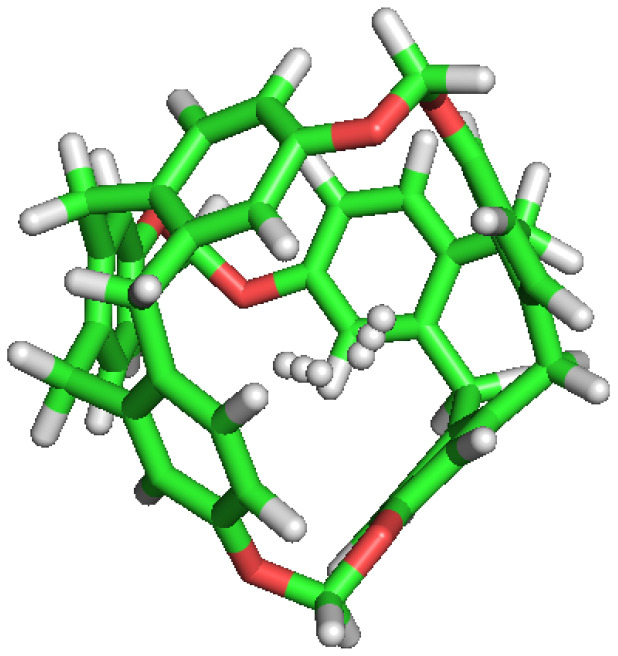
Double occupation of the same cavity by two different H${}_{2}$ molecules.

**Figure 5 nanomaterials-13-00409-f005:**
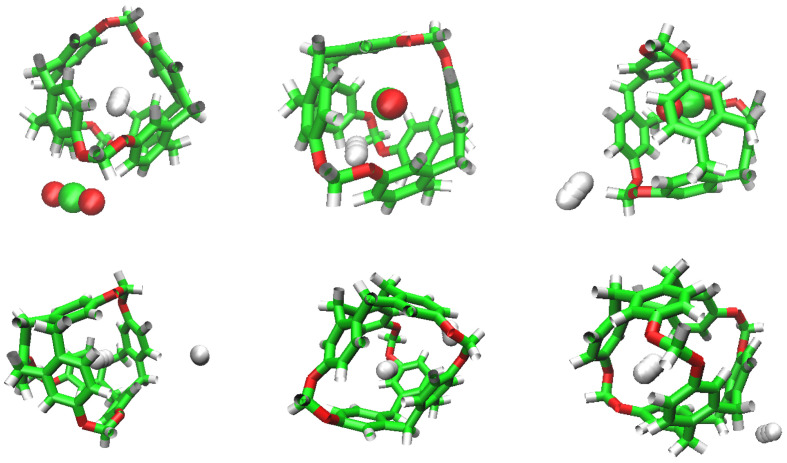
Sequence of H${}_{2}$ replacement by CO${}_{2}$ (above) and by another H${}_{2}$ (below).

**Figure 6 nanomaterials-13-00409-f006:**
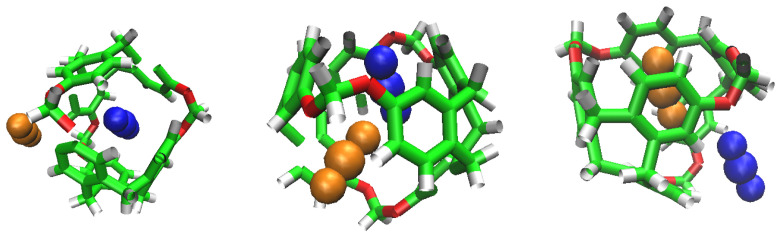
CO${}_{2}$ replacement by another CO${}_{2}$. Beginning, Half and Final frames.

**Figure 7 nanomaterials-13-00409-f007:**
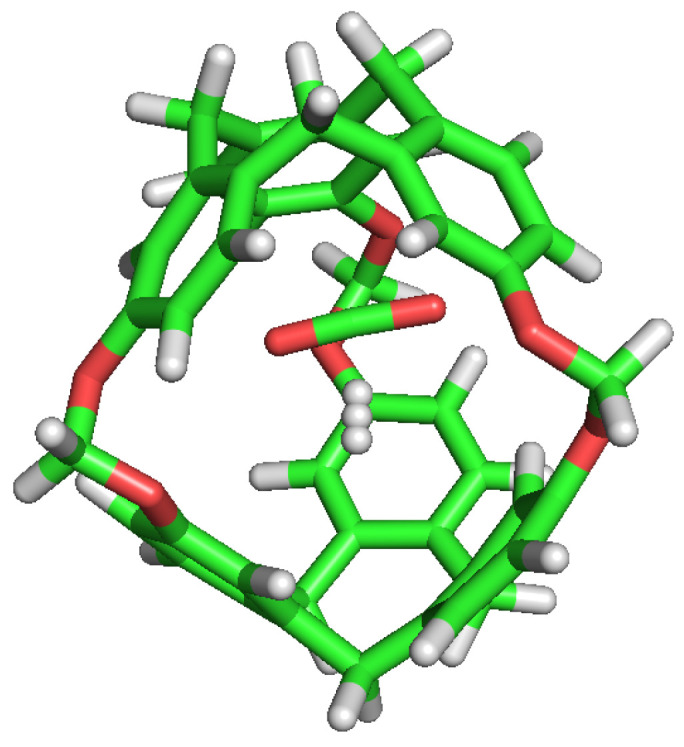
CO${}_{2}$ and H${}_{2}$ sharing the same cavity during a ST2 series simulation.

**Figure 8 nanomaterials-13-00409-f008:**
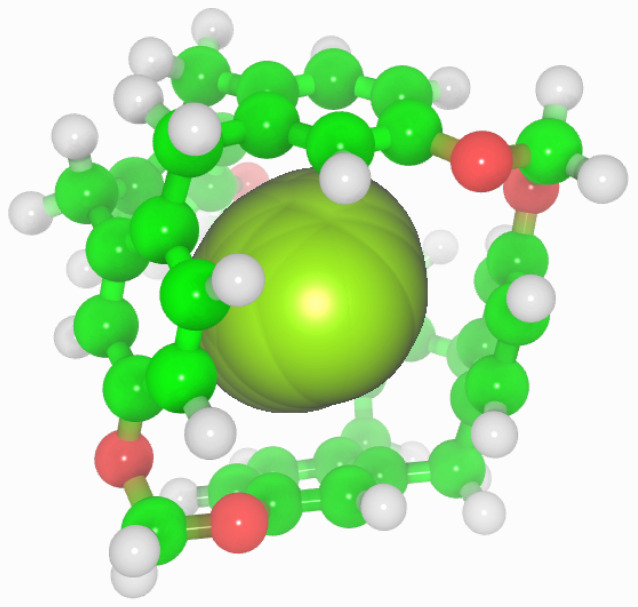
Representation of the accessible volume of the cavity calculation of C-111.

**Figure 9 nanomaterials-13-00409-f009:**
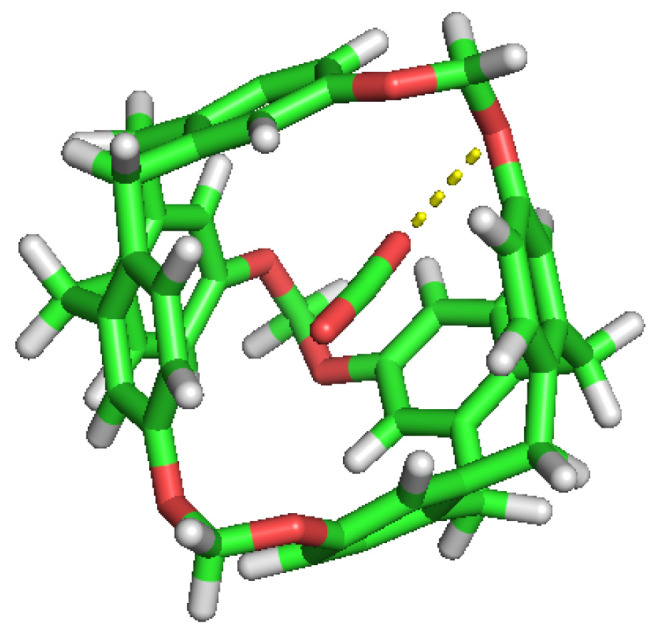
CO${}_{2}$ guest molecule interaction with the ether group inside the cavity.

**Figure 10 nanomaterials-13-00409-f010:**
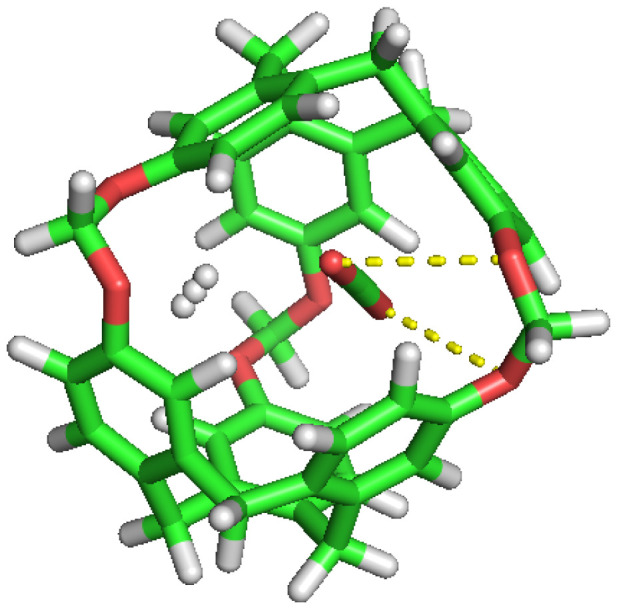
CO${}_{2}$ and H${}_{2}$ inside the cavity, showing their polar interactions.

**Figure 11 nanomaterials-13-00409-f011:**
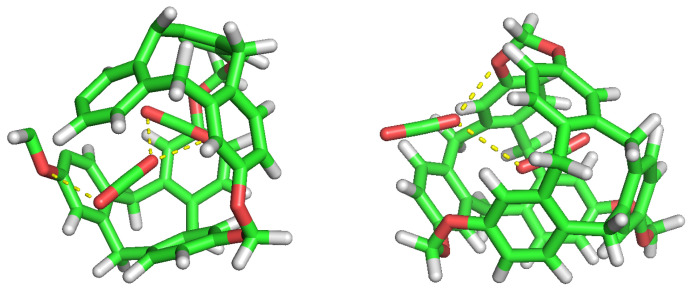
Double CO${}_{2}$ occupancy within a cavity showing their interactions.

**Table 1 nanomaterials-13-00409-t001:** Simulation results for the ST runs (300 K and 1 bar).

Standard					
**Simulations**	**CO** ${}_{2}$	**H** ${}_{2}$	**Free Cavities**	**CO**${}_{2}$/**H**${}_{2}$**In**	**CO**${}_{2}$/**H**${}_{2}$**Out**
S1ST					
Beginning	0	3	4	-	-
Half	3	3	2	3/3	0/3
End	4	1	2	1/1	0/3
S2ST					
Beginning	1	0	6	-	-
Half	1	3	3	0/3	0/0
End	1	3	3	0/2	0/2
S3ST					
Beginning	0	0	7	-	-
Half	2	3	2	2/3	0/0
End	2	3	2	2/3	0/3
S4ST					
Beginning	0	0	7	-	-
Half	1	3	3	1/3	0/0
End	1	3	3	0/2	0/2

**Table 2 nanomaterials-13-00409-t002:** Simulations for the LP runs (300 K and 0.005 bar).

Low Pressure					
**Simulations**	**CO** ${}_{2}$	**H** ${}_{2}$	**Free Cavities**	**CO** ${}_{2}$ **/H** ${}_{2}$ **In**	**CO** ${}_{2}$ **/H** ${}_{2}$ **Out**
S1LP					
Beginning	4	1	2	-	-
Half	4	3	0	0/3	0/1
End	4	1	2	0/1	0/3
S2LP					
Beginning	1	3	3	-	-
Half	1	4	2	0/4	0/3
End	2	4	1	1/4	1/4
S3LP					
Beginning	2	3	2	-	-
Half	3	3	1	1/2	0/2
End	4	2	1	1/1	0/2
S4LP					
Beginning	1	3	3	-	-
Half	3	2	2	2/2	0/3
End	4	2	1	1/2	0/2

**Table 3 nanomaterials-13-00409-t003:** Simulations for the HT runs (312 K and 1 bar).

High Temperature					
**Simulations**	**CO** ${}_{2}$	**H** ${}_{2}$	**Free Cavities**	**CO** ${}_{2}$ **/H** ${}_{2}$ **In**	**CO** ${}_{2}$ **/H** ${}_{2}$ **Out**
S1HT					
Beginning	4	1	2	-	-
Half	5	2	0	1/2	0/2
End	5	2	0	0/2	0/2
S2HT					
Beginning	1	3	3	-	-
Half	3	4	0	3/4	0/3
End	4	2	1	2/2	1/4
S3HT					
Beginning	2	3	2	-	-
Half	3	3	1	1/2	0/2
End	3	2	2	0/2	0/3
S4HT					
Beginning	1	3	3	-	-
Half	4	2	1	3/2	0/3
End	4	2	1	0/2	0/2

**Table 4 nanomaterials-13-00409-t004:** Simulations for the HP runs (300 K and 100 bar).

High Pressure					
**Simulations**	**CO** ${}_{2}$	**H** ${}_{2}$	**Free Cavities**	**CO** ${}_{2}$ **/H** ${}_{2}$ **In**	**CO** ${}_{2}$ **/H** ${}_{2}$ **Out**
S1HP					
Beginning	4	1	2	-	-
Half	5	1	1	1/1	0/1
End	5	1	1	0/1	0/1
S2HP					
Beginning	1	3	3	-	-
Half	1	3	2	1/3	0/3
End	4	4	0	3/3	0/2
S3HP					
Beginning	2	3	2	-	-
Half	2	3	2	1/3	0/3
End	2	4	1	0/4	0/3
S4HP					
Beginning	1	3	3	-	-
Half	3	1	3	2/1	0/3
End	3	3	1	0/3	0/1

**Table 5 nanomaterials-13-00409-t005:** Simulations for the ST2 runs (300 K and 1 bar).

Standard 2					
**Simulations**	**CO** ${}_{2}$	**H** ${}_{2}$	**Free Cavities**	**CO** ${}_{2}$ **/H** ${}_{2}$ **In**	**CO** ${}_{2}$ **/H** ${}_{2}$ **Out**
S1ST2					
Beginning	4	1	2	-	-
Half	5	2	0	1/2	0/1
End	7	0	0	2/0	0/2
S2ST2					
Beginning	1	3	3	-	-
Half	5	1	1	4/1	0/3
End	5	1	1	0/1	0/1
S3ST2					
Beginning	2	3	2	-	-
Half	4	1	2	2/1	0/3
End	5	2	0	1/2	0/1
S4ST2					
Beginning	1	3	3	-	-
Half	2	4	1	1/4	0/3
End	3	3	1	1/3	0/4

**Table 6 nanomaterials-13-00409-t006:** Estimation of average C-111 cavity inner volume ($\bar{V}$ / Å${}^{3}$), for each set of simulations performed, and the corresponding standard deviations ($\Delta \bar{V}$ / Å${}^{3}$).

Conditions	Nº Cavities	$\bar{V}$/Å${}^{3}$	$\Delta \bar{V}$/Å${}^{3}$
Empty cavities			
ST	9	46	8
HP	3	51	8
LP	4	53	5
HT	3	53	11
CO${}_{2}$ occupied			
ST	27	54	9
HP	16	53	9
LP	13	52	12
HT	15	53	14
H${}_{2}$ occupied			
ST	14	56	10
HP	10	53	11
LP	9	51	5
HT	8	50	11

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
