# Peer review of "Molecular Simulation of CO2 and H2 Encapsulation in a Nanoscale Porous Liquid"

_nanomaterials, 2023, doi:10.3390/nano13030409_

Round 1

Reviewer 1 Report

1) nomencatures and abberviations should be expanded minimum 15 should be added, in this manuscript authors referred only two. I recommend delete that two or expand other parameters 

2) Abstract should be re-written the authors abstract is not matching with the work proposed.

3) The introduction part  should be improved mainly, authors should adress  the following back ground, why the present work is so important need to be addressed

4) Caption of tables should be on top please reframe and follow the international guidelines 

5) Results should be expanded well.

Author Response

Reviewer 1

We would like to thank the reviewer for the detailed revision of our manuscript. Please find the list of all the modifications performed according to the comments proposed:

1) nomenclatures and abbreviations should be expanded minimum 15 should be added, in this manuscript authors referred only two. I recommend delete that two or expand other parameters 

We have updated the abbreviation list of the manuscript, adding the missing references. Now it reads as:

ATB     Automatic Topology Builder

DCM     Dicloromethane

C-111     Cryptophane-111

HP     High Pressure

HT     High Temperature

LP     Low Pressure

MOF     Metal Organic Framework

PDB     Protein Data Bank

PL     Porous Liquid

ST     Standard

TraPPE     Transferable Potentials for Phase Equilibria

2) Abstract should be re-written the authors abstract is not matching with the work proposed.

The redaction of the abstract has been extended, and we hope that it now reflects with more accuracy the scope and main contents of the manuscript. After the rephrasing, it reads as follows:

In this study we analyse from a theoretical perspective the encapsulation of both gaseous H2 and CO2 at different conditions of pressure and temperature in a Type II Porous Liquid, composed by nanometric scale cryptophane-111 molecules dispersed in dichloromethane, using atomistic Molecular Dynamics. Gaseous H2 tends to occupy cryptophane--111's cavities in the early stages of the simulation, however, a remarkably greater selectivity of CO2 adsorption can be seen in the course of the simulation. Calculations were performed at ambient conditions first, and then varying temperature and pressure, obtaining some insight about the different adsorption found in each case. An evaluation of the host molecule cavities accessible volume was also performed, based on the guest that occupies the pore. Finally, a discussion between the different intermolecular host-guest interactions is presented, justifying the different selectivity obtained in the molecular simulation calculations. From the results obtained, the feasibility of a renewable separation and storage method for CO2 using these nanometric scale Porous Liquids is pointed out.

3) The introduction part  should be improved mainly, authors should address  the following back ground, why the present work is so important need to be addressed

We have added a new paragraph in the Introduction section, where we have tried to connect the overview presented on Porous Liquid research with the objectives envisaged in this work, consisting in the development of a detailed fully atomistic Molecular Dynamics approach to analyze the selectivity of a type II Porous liquid on a mixture containing CO2 and H2. This new paragraph reads as follows:

The mild equilibrium thermodynamic conditions of PLs in general, and that of the studied case studied here in particular, and the remarkable selectivity in the encapsulation of different guest molecules have opened innovative perspectives in their used as separation and capture media for different gases. This includes for instance the perspective of separating greenhouse gases from industrial flue gases, playing a role in the policies against global warming and climate change. The results presented in this work using Molecular Dynamics for the simplest molecule of the cryptophane family open new perspectives of further analysis of more complex molecular geometries, including even the possibility of adding grafted functional groups to enhance selectivity. The theoretical approach used in this work shows remarkable ability, as it will be demonstrated in the following, to capture the physico-chemical subtleties of this PL selective adsorption process.

4) Caption of tables should be on top please reframe and follow the international guidelines 

Table captions have now been placed on top, we apologize for this mistake of the previous version, caused by our misinterpretation of the journal Latex template, which was used to write and edit the manuscript.

5) Results should be expanded well.

We do not actually understand the meaning of this comment. The results obtained from our molecular simulations have been explained with detail in the manuscript, providing not only averages for the uptake values obtained, but also describing quantitatively each replica calculation performed in order to ensure reproducibility in the calculations and thus reliability of the conclusions. The competitive effect between the adsorption of the different gases considered, CO2 and H2, has been explained also in a comprehensive manner through the analysis of the mutual intermolecular interactions, trying to provide insight into the capture phenomenon.

Reviewer 2 Report

In the submitted manuscript to Nanomaterials entitled “Molecular Simulation of CO2 and H2 encapsulation in nanoscale Porous Liquid” three types of porous liquids are introduced and discussed as a new class of adsorbent material for capture/storage applications. The case study of DCM as solvent and cryptophane has been employed as a cage molecule. The manuscript is written OK supporting neat simulations, however, the areas that require clarification are given below.

·         To this reviewer, the level of originality is unclear or otherwise not well explained in the last part of the introduction.

·         In the abstract, what does the phrase “can be appreciated” mean/refer to?

·         Is there any chemical structure for liquid solvent to be included in Figure 1? Does this solvent have affinity towards molecules like CH4 for gas encapsulation.

·         Yes, porous liquids are a good compromise between the porous solid materials (for absorption) and the liquids, which are easier to handle at the larger scale. However, this has not been well explained in the introduction section. The porous solid materials / liquids reported in the domain for storage (such as doi.org/10.1002/cplu.202200126; doi.org/10.1016/j.ceja.2021.100158) must be made and address the imperative necessity of these porous liquids. Otherwise, it may be harder for the readers to understand.

·         PLs synthesis often leads to low yield and considering the harsh production conditions, they are not very industrially attractive, justify.

·         Do the storage capacity correlate to the gas kinetic sizes, in terms of cage and shape of the gas molecule?

·         What is PLs outlook in CO2 capture and gas separation?

·         How the interfacial behavior and transport dynamics have been characterized through simulation?

·         All table captions must be above the table.

Author Response

Reviewer 2:

We would like to acknowledge the thorough revision provided for our manuscript. We have tried to address all comments presented, which have been very useful to improve the original version submitted. In the following we provide itemized answers for each point raised:

 - To this reviewer, the level of originality is unclear or otherwise not well explained in the last part of the introduction.

A new paragraph has been included at the endo of the introduction with the objective to make clear the originality of the study presented in the manuscript, this new paragraph reads as follows:

The mild equilibrium thermodynamic conditions of PLs in general, and that of the studied case studied here in particular, and the remarkable selectivity in the encapsulation of different guest molecules have opened innovative perspectives in their used as separation and capture media for different gases. This includes for instance the perspective of separating greenhouse gases from industrial flue gases, playing a role in the policies against global warming and climate change. The results presented in this work using Molecular Dynamics for the simplest molecule of the cryptophane family open new perspectives of further analysis of more complex molecular geometries, including even the possibility of adding grafted functional groups to enhance selectivity. The theoretical approach used in this work shows remarkable ability, as it will be demonstrated in the following, to capture the physico-chemical subtleties of this PL selective adsorption process.

 -In the abstract, what does the phrase “can be appreciated” mean/refer to?

The Abstract has been rephrased in order to provide some more detail about the content of the manuscript, and also to clarify the meaning of the expression pointed out by the referee. This sentence has been modified and now is:

Calculations were performed at ambient conditions first, and then varying temperature and pressure, obtaining some insight about the different adsorption found in each case

 -Is there any chemical structure for liquid solvent to be included in Figure 1? Does this solvent have affinity towards molecules like CH4 for gas encapsulation. 

Actually, there is no structure for liquid solvent to be added in Figure 1. The individual molecular models for all the species considered in the study are later depicted in Figure 3, for a better comprehension of the approach used. From results obtained in previous simulation works of our group concerning methane adsorption, its dispersive and non-polar nature allows to estimate that no noticeable interaction should appear with the C-111 molecules leading to an enhanced adsorption. For the case of methane, enclathration results more often from meeting a matching porous geometry with appropriate cavity sizes.

 -Yes, porous liquids are a good compromise between the porous solid materials (for absorption) and the liquids, which are easier to handle at the larger scale. However, this has not been well explained in the introduction section. The porous solid materials / liquids reported in the domain for storage (such as doi.org/10.1002/cplu.202200126;doi.org/10.1016/j.ceja.2021.100158) must be made and address the imperative necessity of these porous liquids. Otherwise, it may be harder for the readers to understand.

We acknowledhe this suggestion from the referee, and we have included the proposed argument in the text, adding also some recent review articles concerning the latest developments in porous materials synthesis and characterization:

In this sense, porous liquids represent a good compromise between the great variety of known solid porous materials used for gas selective adsorption, and liquids, which are easier to store and handle at larger scale. Recent progress in the development and characterisation of porous materials has been reviewed in several excellent compilations

 -PLs synthesis often leads to low yield and considering the harsh production conditions, they are not very industrially attractive, justify.

We agree with this argument raised by the referee, but it is related with the novelty of these materials. The situation was the same when the study of other types of porous materials was initiated, as the cases of zeolites and MOFs. The development in synthesis techniques followed the initial research that demonstrated the potential applications, and the common opinion is that this is the case also for PLs. The question of low yield is also a matter of synthesis tuning, as the inner cavity of these systems can be decorated using surface chemistry techniques to graft selective interacting groups increasing overall adsorbent ability. Research for porous liquids is in its earliest stages right now, but the results are promising and the activity related published in many high level scientific journals demonstrates their promising features.

 -Do the storage capacity correlate to the gas kinetic sizes, in terms of cage and shape of the gas molecule?

Yes indeed, this is one of the key features pointed out from our conclusions. As in the case of zeolites or MOFs, the tunability of the molecules suspended in these porous fluids offers a wide range of synthesis options to fit the desired adsorbed gas molecules kinetic size with dedicated porous geometries. In this sense, Molecular Dynamics is an especially suitable approach if compared with thermodynamic adsorption theories or even Monte Carlo simulation. The reason is that MD shows explicitly not only the existence of an equilibrium adsorbed state, but also the kinetic aspects of accessibility of these equilibrium states, that are known to be hardly accessible in many adsorption phenomena due to steric constraints, for instance. The time evolution of these equilibrium states is also pointed out through the direct evaluation of the competence between CO2 and H2 as guest molecules inside the C-111 cavities in our calculations.

 -What is PLs outlook in CO2 capture and gas separation?

As described in the conclusions section, these C-111 based PLs are promising systems due to the CO2 selectivity, and also to the high CO2 uptake with a very high occupancy ratio of the available cavities. The present study is exploratory and can be considered as a first step, as further detailed analysis must follow including questions as system size scaling, possibility of C-111 clogging, and other negative effects for the envisaged application that have not been considered in this case.

 -How the interfacial behavior and transport dynamics have been characterized through simulation?

Interfacial behaviour has not been characterized explicitly. The referee must be aware that the computation of interfacial behaviour in solid-fluid systems (as ice or hydrates for instance) is a matter of debate nowadays with new techniques and challenging developments being proposed over the last few years, so this was unfortunately beyond the scope of the present work. Transport dynamics was considered only as described, by characterizing the transition between empty C-111 and loaded PLs, but not taking into account any transport coefficient in particular as objective of our simulations. These two ideas proposed are open questions in the field of PLs, hopefully to be addressed in future works.

 -All table captions must be above the table.

This has been changed in the manuscript, we made a mistake in the original version with the journal latex template options, we apologize for this.

Reviewer 3 Report

The authors presented a very interesting Molecular Simulation of CO2 and H2 encapsulation in nanoscale Porous Liquid

The abstract is to be extended.

The novelty of the paper is to be clearly stated.

The computational method is to be detailed.

The assumptions used in the simulations are to be presented.

The paper is to be checked against misprint and grammatical mistakes.

The discussion is to be improved by adding physical interpretations.

Author Response

Reviewer 3

- The authors presented a very interesting Molecular Simulation of CO2 and H2 encapsulation in nanoscale Porous Liquid

We sincerely acknowledge the overall positive evaluation that the referee has expressed concerning our manuscript

- The abstract is to be extended.

According to the referee comment the abstract has been extended in order to provide a more accurate description of the paper objectives and contents. In the current version it reads as follows:

In this study we analyse from a theoretical perspective the encapsulation of both gaseous H2 and CO2 at different conditions of pressure and temperature in a Type II Porous Liquid, composed by nanometric scale cryptophane-111 molecules dispersed in dichloromethane, using atomistic Molecular Dynamics. Gaseous H2 tends to occupy cryptophane--111's cavities in the early stages of the simulation, however, a remarkably greater selectivity of CO2 adsorption can be seen in the course of the simulation. Calculations were performed at ambient conditions first, and then varying temperature and pressure, obtaining some insight about the different adsorption found in each case. An evaluation of the host molecule cavities accessible volume was also performed, based on the guest that occupies the pore. Finally, a discussion between the different intermolecular host-guest interactions is presented, justifying the different selectivity obtained in the molecular simulation calculations. From the results obtained, the feasibility of a renewable separation and storage method for CO2 using these nanometric scale Porous Liquids is pointed out.

- The novelty of the paper is to be clearly stated.

A new paragraph has been added at the end of the Introduction section in order to clarify the novelty of the contribution presented in this manuscript. This is the new paragraph included:

The mild equilibrium thermodynamic conditions of PLs in general, and that of the studied case studied here in particular, and the remarkable selectivity in the encapsulation of different guest molecules have opened innovative perspectives in their used as separation and capture media for different gases. This includes for instance the perspective of separating greenhouse gases from industrial flue gases, playing a role in the policies against global warming and climate change. The results presented in this work using Molecular Dynamics for the simplest molecule of the cryptophane family open new perspectives of further analysis of more complex molecular geometries, including even the possibility of adding grafted functional groups to enhance selectivity. The theoretical approach used in this work shows remarkable ability, as it will be demonstrated in the following, to capture the physico-chemical subtleties of this PL selective adsorption process.

- The computational method is to be detailed.

All the molecular simulation models, details of the setup, and relevant variables have been declared in the text. This means that the performed simulation could be repeated by any interested reader by cloning exactly the system described. We could provide more details about the basic foundations of the simulation (simulation ensemble, integration algorithms, choice of long range correction scheme, thermostat and barostat applied), but these details can be found in any molecular simulation reference manual and including them would only contribute to a unnecessary lengthy manuscript, distracting attraction from simulation results and PL behaviour.

- The assumptions used in the simulations are to be presented.

The molecular simulation technique used is declared, and no further assumption is used. Actually, when planning the study we decided on purpose  to use the most detailed molecular modelling scale, with a fully atomistic description of all molecules present. The importance of this decision has to be taken into account, as it increases computing times by an order of magnitude if compared with simpler approaches that would be accessible, as a coarse graining approach. Nevertheless, the lack of studies concerning this type of systems recommends avoiding the integration of any degree of freedom, in order not to miss any effect caused by the chemical complexity of the system. Thus, no spurious reduction scheme has been applied, and the approach used is as detailed and transparent

- The paper is to be checked against misprint and grammatical mistakes.

We acknowledge the referee for this comment, the manuscript has been revised and several typos and grammar erros have been fixed.

- The discussion is to be improved by adding physical interpretations.

In the conclusions section we decided to stick strictly to our results, avoiding any speculative interpretation. Perhaps some more physical discussion might be drawn from our results, but frankly speaking this would mean assuming risks by using arguments not fully supported by our results. In the present state, we are fully confident with the interpretation and conclusions provided, and we prefer to obtain further results concerning this or similar PLs before going further in the system complex behaviour interpretation.